# Multiview Spatial-Temporal Meta-Learning for Multivariate Time Series Forecasting

**DOI:** 10.3390/s24144473

**Published:** 2024-07-10

**Authors:** Liang Zhang, Jianping Zhu, Bo Jin, Xiaopeng Wei

**Affiliations:** 1School of Computer Science and Technology, Dalian University of Technology, Dalian 116024, China; liangzhang@dlut.edu.cn (L.Z.); zhujp@mail.dlut.edu.cn (J.Z.); xpwei@dlut.edu.cn (X.W.); 2School of Innovation and Entrepreneurship, Dalian University of Technology, Dalian 116024, China

**Keywords:** multivariate time series prediction, spatial and temporal meta-learning, graph neural network, dynamic relations

## Abstract

Multivariate time series modeling has been essential in sensor-based data mining tasks. However, capturing complex dynamics caused by intra-variable (temporal) and inter-variable (spatial) relationships while simultaneously taking into account evolving data distributions is a non-trivial task, which faces accumulated computational overhead and multiple temporal patterns or distribution modes. Most existing methods focus on the former direction without adaptive task-specific learning ability. To this end, we developed a holistic spatial-temporal meta-learning probabilistic inference framework, entitled ST-MeLaPI, for the efficient and versatile learning of complex dynamics. Specifically, first, a multivariate relationship recognition module is utilized to learn task-specific inter-variable dependencies. Then, a multiview meta-learning and probabilistic inference strategy was designed to learn shared parameters while enabling the fast and flexible learning of task-specific parameters for different batches. At the core are spatial dependency-oriented and temporal pattern-oriented meta-learning approximate probabilistic inference modules, which can quickly adapt to changing environments via stochastic neurons at each timestamp. Finally, a gated aggregation scheme is leveraged to realize appropriate information selection for the generative style prediction. We benchmarked our approach against state-of-the-art methods with real-world data. The experimental results demonstrate the superiority of our approach over the baselines.

## 1. Introduction

Advanced data acquisition technologies lead to the real-time collection of sequential signals, thus inducing widespread data analysis and downstream applications, such as electricity forecasting [1,2], traffic flow prediction [3,4,5], inventory control [6], healthcare management [7], etc. In essence, due to inherent connections among multiple univariate time series, most of the aforementioned scenarios can be regarded as multivariate time series modeling tasks, which play a key role in the decision-making process of digital management. For instance, the sales data-driven prediction of product supply for chain stores can optimize inventory management, helping save millions of dollars in each year. However, capturing the complex temporal dynamics of the multivariate time series data is a very challenging task.

First, there exist complex intra- and inter-variable relations. For each variable, temporal dependencies might be entangled with different evolving patterns, e.g., various trends and seasonality [2]. Existing autoregressive methods, filtering methods, and deep forecasting models via recurrent connections, temporal attention, and causal convolution can only capture limited modalities. In addition, there exist interdependent and even causal relations [8,9], i.e., each variable depends not only on its past value but also on other variables. Nonetheless, these relations are not always available as prior knowledge. For example, pairwise relations of different stations might not be given in the electricity forecasting task. Hence, it is mission-critical to automatically extract underlying relation structures and then develop structure-enhanced temporal modeling techniques, especially about how to fuse temporal and spatial information. Recent advances [8,9,10] seek to couple graph convolution with temporal recurrent processing at each timestamp. Instead, inspired by the effectiveness of separate autoregressive models [11] and graph structures [12,13] and by the disentangled [14] and fine-grained multiview learning [15] in time series modeling, we argue that the multiview fine-grained fusion of temporal view- and spatial view-specific information are essential to make up for the information loss brought by tightly coupled schemes.

Second, mainstream learning paradigms assume the distribution consistency of the training and inference stages. Nonetheless, in most multivariate time series forecasting tasks, temporal patterns keep evolving [16] and can be significantly different (e.g., temporal burst, multivariate relationship changes), such that it can be regarded as time series modeling in diverse learning environments. One promising direction is the application of meta-learning strategies, e.g., MAML [17], in time series prediction tasks [18,19], which captures meta knowledge shared by different tasks for fast adaption in new few-shot learning tasks. For example, diverse spatio-temporal purchasing patterns brought by heterogeneous external factors require high-capacity multiple-task-aware meta-learning models [6] that can adapt to different learning environments via knowledge transfer. Despite some advances in spatio-temporal meta-learning [3,6,19], it is still unclear how to meta-fuse temporal and spatial information for better task-transfer performances. Furthermore, the evolving complex inter-variable relationships and intra-variable temporal information add further challenges to the fast-changing learning tasks.

Third, a fast adaptation capability to new or recurring patterns while being aware of the diversity of distributions is crucial in most multivariate time series forecasting tasks. For instance, the sudden change in electricity production might be caused by a change in the natural environment or mechanical problems, and a model should be able to capture such diversity while quickly adapting to current data distributions. While some recent works have successfully improved learning efficiency via either time series temporal modeling itself [1,2] or meta-learning schemes [6] and capture diversity via stochastic neurons in conditional generative models [20], the inherent multi-step nature of time series present challenges about how to realize position-specific diversities.

To tackle all the above challenges, we propose a holistic spatial-temporal meta-learning probabilistic inference framework, entitled ST-MeLaPI, for the efficient and versatile learning of complex dynamics of multivariate time series data. Unlike previous meta-learning scenarios where different time series learning tasks have clear boundaries, we focus on a general setting in which only time series data are provided without task specification. Recent progress in online time series forecasting [16] and continuous spatial-temporal meta-learning [21] inspired us to consider that samples in local consecutive time slots can be regarded as a specific learning task given the modality-rich and time-evolving nature of multivariate time series data. We chose the meta-learning framework VERSA [22] because it realizes multi-task probabilistic learning across all tasks through an amortization network that can directly take support datasets (with any number of shots, i.e., flexibility) as inputs, and outputs a distribution (rather than fixed value, i.e., versatility) over task-specific parameters in a single forward pass (i.e., rapid inference). Hence, it is more appropriate for capturing diverse temporal patterns spanning different periods of time series data. In essence, ST-MeLaPI can be regarded as a sequential structure derivation of VERSA.

More concretely, we first formalize diverse learning tasks by dividing the original data into different batches. For each task, a multivariate relationship identification module (MRI) [9] is leveraged to learn the task-specific inter-variable dependencies. Then, a multiview meta-learning and probabilistic inference (MV-MeLaPI) strategy is developed. Specifically, first, given the support set and query set in a task, a multiview feature encoder (MFE) captures spatial dependency- and temporal pattern-induced view-specific information, respectively. Second, a multiview position-aware network (MV-PAN) was designed to map the spatial dependency- and temporal pattern-induced information (after concatenation with prediction window data) of the support set information into stochastic input at each timestamp, respectively. Third, a cross-view gated generator (CV-GAG) is leveraged to realize cross-view position-aware fusion and generative style predictions at one forward operation.

To sum up, our main contributions are the following:We propose a novel framework based on meta-learning approximate probabilistic inference, entitled ST-MeLaPI, for the efficient and versatile learning of multivariate time series data. ST-MeLaPI regards that each short time span serves as the dataset for a specific task and then leverages a pairwise variable relationship learner (MRI) to capture task-specific inter-variable dependencies, hence enabling a spatial dependency- and temporal pattern-induced approximate posterior for one forward all-timestamp prediction (MV-MeLaPI).We developed a multiview position-aware meta-learning and inference strategy: MV-MeLaPI. It enables fast and versatile learning via position-aware stochastic neurons. Spatial dependency- and temporal pattern-induced view-specific stochastic inputs from the support set are generated. Then, a cross-view information concatenation of stochastic inputs and encoded (view-specific) query set was designed to the approximate posterior predictive distribution, which realizes multiview meta-probabilistic inference learning at each timestamp.ST-MeLaPI can be used for both long-sequence and short-sequence time-series forecasting tasks. In this study, we conducted several experiments with real-world data. The extensive experimental results show improved learning performances.

## 2. Related Work

Multivariate time series forecasting techniques have attracted much attention in various application areas. Mainstream techniques can be divided into three categories: (1)Classic statistical tools. Typical approaches, such as ARIMA [23], Kalman Filters [24], and linear state-space models [25], are essentially linear, thus only capturing low-order temporal dynamics. Moreover, they cannot benefit from shared temporal patterns in the whole dataset.(2)Deep sequential neural networks. Nonlinear recurrent neural network (RNN) models [26] have shown advantages in capturing high-order complex dynamics. For instance, LSTNet [27] combines the convolutional neural network with recurrent skip connections to extract short-term and long-term temporal patterns. DeepAR [11] performs probabilistic prediction through the combination of an autoregressive method and RNN. Transformer-based models [1,2,28,29,30] utilize self-attention mechanisms [31] to capture the long-range dependencies for prediction. In addition, another popular and competitive architecture is the causal convolution or temporal convolutional network (TCN) [32]. For example, DeepGLO [33] combines a global matrix factorization model with temporal convolutional networks. Even though recent state-of-the-art decomposition-based models [2,34,35,36] achieve superior performances, they are mainly designed for capturing the seasonal or trend-cyclical patterns in time series, which is far from satisfactory due to the versatile patterns of time series, especially considering the evolving nature of the data. In light of this, some works adopt meta-learning strategies [18] to realize quick task adaption, or choose multitask learning [37] for adaptive learning and forecasting in non-stationary data contexts.(3)Multivariate relationship-aware techniques. The underlying inter-variable dependencies, e.g., correlation and causation, have shown promise in enhancing the prediction performance [9]. Therefore, recent advances focus on spatio-temporal learning; in particular, traffic forecasting models [3,4,10,38,39,40,41,42] consider spatial connections through the combination of graph neural networks (GNNs) [43] and recurrent neural networks, thus forecasting all time series simultaneously. However, the aforementioned methods assume the spatial connections of roads as prior knowledge, which is often not available in most multivariate time series forecasting tasks. Therefore, automatic inter-variable dependency structure learning has been integrated into temporal dynamics modeling [8,9,12,44]. For instance, NRI [8] uses an autoencoder approach that learns different structures given different multivariate time series inputs. GTS [9] first learns a probabilistic graph for the whole time series and then combines the graph neural network and temporal recurrent processing to model inter-variable structure-aware temporal dynamics. Nevertheless, these models either only learn a single structure given one set of *K* series spanning a whole time window [9] or cannot capture the diverse temporal patterns with fixed model parameters despite dynamic structures [8]. Recently, ML-PIP [6,22] was proposed, adopting a multitask meta-learning framework to perform approximate probabilistic inference under low-resource learning environments. However, on the one hand, ML-PIP assumes that spatial information is given and divides all data into different learning tasks according to spatial and temporal information simultaneously; on the other hand, it does not consider the multivariate time series prediction problem, much less a position-aware multiview meta-learning strategy. Hence, it is very different from our learning tasks and proposed model. But it provides us the insight that each short time period can be naturally regarded as a specific learning task given the modality-rich and time-evolving nature of multivariate time series, thus realizing data-efficient and versatile learning. However, the inherent multi-step nature of time series present challenges about how to realize the time-specific diversity. In particular, the effectiveness of sequential dynamics-only [1,11,27,33], graph structure-only [12,13] and entangled spatio-temporal modeling [9,44] motivated us to design multivariate relationships and temporal dynamics in a more sophisticated way. For example, Qu et al. [13] proposed a sparse connectivity network to identify temporal dynamics using an adaptive regularization approach; Informer [1] shows the power of self-attention in sequential modeling. In particular, both temporal and contextual information [15] enhance the representation learning of multivariate time series data, strengthening the necessities of the fine-grained multiview fusion of temporal and spatial information.

## 3. Methodology

In this section, we provide an explicit problem definition and then describe the architecture of our proposed framework, ST-MeLaPI (Figure 1), which consists of Multivariate Relationship Identification (MRI) and Multiview Meta-Learning and Probabilistic Inference (MV-MeLaPI).

### 3.1. Problem Definition

A multivariate time series forecasting task aims at modeling the conditional distribution $P(Y^{\left(i\right)}|X^{\left(i\right)})$ of the *i*th dynamic system with a set of *K* series, where $X^{\left(i\right)}\in R^{T_{x}\times K}$ represents the conditioning range (past $T_{x}$ timestamps) data, and $Y^{\left(i\right)}\in R^{T_{y}\times K}$ represents the prediction range (future $T_{y}$ timestamps) data. Note that we assume that the *K* series have already been aligned. In particular, our key motivation is that samples in a short time span serve as a specific learning task. Hence, we split the whole time series data into different batches. Each batch consisting of $X\in R^{B\times T_{x}\times K}$ and $Y\in R^{B\times T_{y}\times K}$ contains *N* support sets and *M* query sets with B=N+M. In our work, we let $X_{b}=\left\{x_{b,1},\cdots ,x_{b,T_{x}}|x_{b,t}\in R^{K}\right\}$ and $Y_{b}=\left\{y_{b,1},\cdots ,y_{b,T_{y}}|y_{b,t}\in R^{K}\right\}$ represent the input and output of the *b*th sample in the batch, respectively. By learning from different short time periods in the past, our model aims at capturing shared patterns (or parameters) via knowledge transfer while realizing fast and versatile learning and inference for future predictions via stochastic neurons at each timestamp.

### 3.2. Multivariate Relationship Identification

The Multivariate Relationship Identification module (MRI) aims at improving the forecasting performance by exploiting inter-variable relationships. In our work, we implemented MRI by following GTS [9]. It consists of an univariate time series feature extractor, an inter-variable linkage predictor, and a differentiable structure sampler based on the Gumbel-Softmax reparameterization trick.

To be specific, first, a univariate feature extractor maps each variable $X^{i}$ to a feature vector $c^{i}$:(1)$$c^{i}=W_{c}\left(conv1d\left(X^{i}\right)\right)+b_{c},$$
where $W_{c}$ and $b_{c}$ are trainable parameters, and conv1d is a one-dimensional convolutional layer that captures a global intra-variable temporal pattern. Then, the link predictor yields a link probability $\delta_{ij}\in \left[0,1\right]$ with Bernoulli distribution for each pair of feature vectors $c^{i}$ and $c^{j}$:(2)$$\delta_{ij}=\sigma (MLP(c^{i}\left|\right|c^{j})),$$
where || denotes the concatenation operation, MLP represents a multi-layer perceptron, and σ represents a sigmoid activation function. Finally, the Gumbel reparameterization technique [45] is leveraged to solve the differentiability problem brought by the sampling operation, while generating global structure diversity. The final multivariate relationship structure *A* is learned as follows:(3)$$A_{ij}=\sigma \left(\left(\log\left(\delta_{ij}/\left(1-\delta_{ij}\right)\right)+\left(g_{ij}^{1}-g_{ij}^{2}\right)\right)/\tau \right),$$
where $g_{ij}^{1},g_{ij}^{2}∼Gumbel\left(0,1\right)$ for all *i*, *j*. When the temperature τ tends to 0, $\delta_{ij}$ is used as the probability of $A_{ij}=1$.

Since each task spans short time periods, the inter-variable relationship can be regarded as invariant in each task. Therefore, each sample shares the same structure, which is generated by inputting the whole conditioning range data in the support set into the MRI module. Moreover, other pairwise relationship structure learning approaches can be used as well, e.g., NRI [8] and MTPool [44]. We followed GTS since the end-to-end structure learning and diverse structure sampling strategy are more intuitive for flexible and efficient learning in multivariate time series prediction tasks.

### 3.3. Multiview Meta-Learning and Probabilistic Inference

Based on the task-specific multivariate relationship structure *A*, we can conduct inter-variable relationship-aware temporal recurrent processing. Different from coupled combination [9,10,39,44], we seek a fine-grained fusion of view-specific information. At the core is a multiview position-aware spatial-temporal meta-learning strategy MV-MeLaPI by following VERSA [22]. It uses meta-learning to capture shared information between tasks, and enables fast and versatile learning through an amortization network. As a temporal variant of VERSA, MV-MeLaPI realizes efficient and versatile model learning and approximate prediction via stochastic neurons at each timestamp. Fundamentally, it is composed of a Multiview Feature Encoder (MFE), a Multiview Position-aware Amortization Network (MV-PAN), and a Cross-view Gated Generator (CV-GAG). Both MFE and CV-GAG are shared parameters (i.e., θ in VERSA) that are common to all tasks, while MV-PAN (i.e., ϕ in VERSA) helps generate task-specific stochastic inputs ψ (parameters) for a regression (classification) task.

Note that global auxiliary information, such as hierarchical time stamps (week, month, and year) and agnostic time stamps (holidays, events), offers guidance in capturing temporal dynamics [1]. Therefore, we incorporate global auxiliary information through a linear mapping operation and concatenation with the original multivariate time series data. Finally, we obtain the input $Z\in R^{B\times T\times K\times Fe}$ of the MFE through a linear mapping:(4)$$Z=W_{H}(X_{en}\left||Time\right)+b_{H},$$
where $Time\in R^{B\times T\times K\times 1}$; $W_{H}$ and $b_{H}$ are trainable parameters; and Fe represents the dimension of the initial embedding for each variable. We take $X_{en}=Concat\left(X,X_{0}\right)\in R^{B\times T\times K\times 1}$ as input data, where $X_{0}\in R^{B\times T_{y}\times K}$ is a placeholder (set as 0) for the target sequence, and $T=T_{x}+T_{y}$.

#### 3.3.1. Multiview Feature Encoder

The Multiview Feature Encoder (MFE) was designed to extract spatial dependency- and temporal pattern-induced view-specific features, leading to comprehensive information utilization. Specifically, it consists of a coarse-grained graph convolutional network (C-GCN) and a fine-grained Gated Recurrent Unit (F-GRU).

**Coarse-grained Graph Convolutional Network**. The C-GCN was designed to capture the spatial dependency-induced temporal dynamics. Note that we call it “coarse-grained” since it only captures inter-variable relationship-enhanced feature learning at each timestamp without modeling sequential temporal transitions. In effect, it learns a structure-enhanced feature embedding for each variable through a graph convolution. Specifically, based on the learned batch-specific (task-specific) adjacency matrix *A*, we used the GCN to obtain an updated representation for each variable, formalizing $Z^{c}$:(5)$$Z^{c}=\sigma \left({\tilde{D}}^{-\frac{1}{2}}\tilde{A}{\tilde{D}}^{-\frac{1}{2}}ZW_{C}\right),$$
where $W_{C}$ is the trainable parameter, $\tilde{A}=A+I$ is the adjacency matrix joining the self-recursive edges, *I* is the unit matrix, and $\tilde{D}$ is the degree matrix of $\tilde{A}$.

**Fine-grained Gated Recurrent Unit**. Most of the existing temporal prediction models rely on discrete snapshots, i.e., the model input is only the recorded data at each time stamp, which fails to explicitly consider the fluctuations of the data within two consecutive time points. To fill this gap, as shown in Figure 2, we propose a fine-grained Gated Recurrent Unit Network (F-GRU), which adds a fine-grained feature-forgetting gate to the traditional GRU model. Each F-GRU cell takes as input the feature vector $z_{t-1}$ with $z_{t}$ of continuous timestamps in $Z=\{z_{1},\cdots ,z_{T}\}$:(6)$$\begin{array}{cc} \; & z_{t-1,t}=W_{h}^{f}(z_{t-1}\left|\right|z_{t})+b_{h}^{f},\\ \; & v_{t}^{f}=\sigma \left(W_{v}^{f}\cdot \left[z_{t-2,t-1},z_{t-1,t}\right]+b_{v}^{f}\right),\\ \; & r_{t}^{f}=\sigma \left(W_{r}^{f}\cdot \left[z_{t-1}^{f},z_{t}\right]+b_{r}^{f}\right),\\ \; & u_{t}^{f}=\sigma \left(W_{u}^{f}\cdot \left[z_{t-1}^{f},z_{t}\right]+b_{u}^{f}\right),\\ \; & {\hat{z}}_{t}^{f}=tanh\left(W^{f}\cdot \left[r_{t}^{f}*z_{t-1}^{f},v_{t}^{f}*z_{t-2,t-1},z_{t-1,t}\right]+b^{f}\right),\\ \; & z_{t}^{f}=\left(1-u_{t}^{f}\right)*z_{t-1}^{f}+u_{t}^{f}*{\hat{z}}_{t}^{f}, \end{array}$$
where $W_{h}^{f}$, $W_{v}^{f}$, $W_{r}^{f}$, $W_{u}^{f}$, $W^{f}$, $b_{h}^{f}$, $b_{v}^{f}$, $b_{r}^{f}$, $b_{u}^{f}$, and $b^{f}$ are all trainable parameters. We first concatenate $z_{t-1}$ with $z_{t}$ as the input of a linear layer to obtain $z_{t-1,t}$, which represents continuous local context information at the *t*th step. The fine-grained forgetting gate $v_{t}^{f}$ controls the degree of forgetting from previous local context information. When updating the memory content ${\hat{z}}_{t}^{f}$ at the *t*th step, our model incorporates the fine-grained features $z_{t-1,t}$, in addition to the reset gate $r_{t}^{f}$ and the fine-grained forgetting gate $v_{t}^{f}$. Finally, at each timestamp, the fine-grained temporal feature vector $z_{t}^{f}$ is generated by the update gate $u_{t}^{f}$.

#### 3.3.2. Multiview Position-Aware Amortization Network

Based on the aforementioned view-specific information, i.e., two view-specific feature embedding for each variable at each timestamp, an MV-PAN was developed to fully exploit shared information among tasks and generate task-specific parameters. Essentially, this amortization network realizes fast and versatile learning through outputting a distribution over task-specific parameters or stochastic inputs in a single forward pass. Specifically, for each task, we have support set $D_{b}^{c/f}={\left\{\left(Z_{b,n}^{c/f},y_{b,n}\right)\right\}}_{n=1}^{N}$, and query set ${\left\{\left({\tilde{Z}}_{b,m}^{c/f},{\tilde{y}}_{b,m}\right)\right\}}_{m=1}^{M}$. As mentioned in VERSA [22], forming a posterior distribution over the task-specific parameters $p(\psi_{b}^{s/d}|D_{b}^{c/f},\theta )$ or stochastic input is usually intractable; hence, approximate posterior distribution $q_{\phi^{s}}\left(\psi^{s}|D^{c}\right)$ and $q_{\phi^{d}}\left(\psi^{d}|D^{f}\right)$ are learned via amortization networks $\phi^{s}$ (spatial) and $\phi^{d}$ (temporal), respectively. In practice, we employ feed forward neural networks for the amortization networks. First, the view-specific spatial-temporal representations of the support set are concatenated with the corresponding prediction window data. Then, an average instance-pooling operation ensures that the network can process any number of training observations while being permutation-invariant. After that, a linear operation helps generate the means and variances of a factorized Gaussian distribution, which approximates the real spatial-temporal posterior distributions over the task-specific stochastic inputs (or task-specific parameter information in a classification task):(7)$$\begin{array}{cc} \; & \mu_{s/d}=W_{\mu_{s/d}}\left(\frac{1}{N}\sum_{i=1}^{N}\left(Z_{i}^{c/f}||y_{i}\right)\right)+b_{\mu_{s/d}},\\ \; & \sigma_{s/d}=W_{\sigma_{s/d}}\left(\frac{1}{N}\sum_{i=1}^{N}\left(Z_{i}^{c/f}||y_{i}\right)\right)+b_{\sigma_{s/d}}, \end{array}$$
where $W_{\mu_{s/d}}$, $W_{\sigma_{s/d}}$, $b_{\mu_{s/d}}$, and $b_{\sigma_{s/d}}$ are trainable parameters. Finally, feature vectors ${\tilde{Z}}^{c/f}$ (generated through the MFE) of the conditioning window from the query set will be concatenated with the above stochastic inputs to form an approximate posterior predictive distribution:(8)$$\begin{array}{cc} q_{\phi }\left(\tilde{y}|D\right) & =\;\int \int \;p(\tilde{y}|\psi^{s},\psi^{d})\\ \; & \;q_{\phi^{s}}\left(\psi^{s}|D^{c}\right)q_{\phi^{d}}\left(\psi^{d}|D^{f}\right)d\psi^{s}d\psi^{d}, \end{array}$$
where the neural network ϕ contains $\phi^{s}$ and $\phi^{d}$. The use of amortized variational inference and neural networks enable fast predictions at test time without traditional second-order derivatives during training. Further details can be seen in VERSA [22]. We will introduce, in detail, how to model $p(\tilde{y}|\psi^{s},\psi^{d})$ and $q_{\phi }\left(\tilde{y}|D\right)$ in the following subsection.

#### 3.3.3. Cross-View Gated Generator

After obtaining posterior distributions $q_{\phi^{s}}\left(\psi^{s}|D^{c}\right)$ and $q_{\phi^{d}}\left(\psi^{d}|D^{f}\right)$ of the spatial view- and temporal view-specific stochastic inputs through the MV-PAN, the Cross-view Gated Generator is used to realize posterior predictions. Here, the cross-view information fusion between view-specific stochastic inputs (from the support set) and encoded conditioning window data (from the query set) enables the query set to capture multiple views of temporal dynamics from the training dataset, while making up for the information utilization loss in each view.

Figure 3 demonstrates how to leverage the shared parameters θ of all tasks and support set to learn task-specific parameter ${\left\{\psi_{b}^{s/d}\right\}}_{b=1}^{B}$ and to perform inference for the query set. To be specific, the spatial relationship-induced parameter $\psi^{s}$ is concatenated to the temporal pattern-induced query set information ${\tilde{Z}}^{f}$ and then sent to a fine-grained graph convolutional network (F-GCN) for predictive predictions. In addition, the temporal pattern-induced parameter $\psi^{d}$ is concatenated to the spatial relationship-induced query set information ${\tilde{Z}}^{c}$ and then sent to a coarse-grained gated recurrent unit network (C-GRU) for further learning.

We use a fusion function Ψ() to complete the initialization of the information:(9)$$\begin{array}{cc} \; & {\tilde{Z}}^{c,\psi }=\Psi \left({\tilde{Z}}^{c},\psi^{d}\right),\\ \; & {\tilde{Z}}^{f,\psi }=\Psi \left({\tilde{Z}}^{f},\psi^{s}\right). \end{array}$$

**Coarse-grained Gated Recurrent Unit.** To broaden the receptive field when learning sequential dynamics, we use an attention mechanism to directly aggregate temporal information from the past w−1 time steps. Then, the reset gate in the Gated Recurrent Unit (GRU) can control the information flow from more steps away. Specifically, as shown in Figure 4,
(10)$$\begin{array}{cc} \; & \alpha_{i}=\frac{\exp({\tilde{z}}_{i}^{c,\psi }\cdot {\tilde{z}}_{t}^{c,\psi ⊤})}{\sum_{j=t-w}^{t-1}\exp({\tilde{z}}_{j}^{c,\psi }\cdot {\tilde{z}}_{t}^{c,\psi ⊤})},\\ \; & f_{t}^{c}=\sum_{i=t-w}^{t-1}\alpha_{i}{\tilde{z}}_{i}^{c,\psi },\\ \; & r_{t}^{c}=\sigma \left(W_{r}^{c}\cdot \left[f_{t}^{c},{\tilde{z}}_{t}^{c,\psi }\right]+b_{r}^{c}\right),\\ \; & u_{t}^{c}=\sigma \left(W_{u}^{c}\cdot \left[h_{t-1}^{c},{\tilde{z}}_{t}^{c,\psi }\right]+b_{u}^{c}\right),\\ \; & {\hat{h}}_{t}^{c}=tanh\left(W^{c}\cdot \left[r_{t}^{c}*f_{t}^{c},{\tilde{z}}_{t}^{c,\psi }\right]+b^{c}\right),\\ \; & h_{t}^{c}=\left(1-u_{t}^{c}\right)*h_{t-1}^{c}+u_{t}^{c}*{\hat{h}}_{t}^{c}, \end{array}$$
where $W_{r}^{c}$, $W_{u}^{c}$, $W^{c}$, $b_{r}^{c}$, $b_{u}^{c}$, and $b^{c}$ are trainable parameters. $f_{t}^{c}$ is a coarse-grained vector generated by the aforementioned attention mechanism, and $r_{t}^{c}$ is a coarse-grained reset gate. The coarse-grained feature vector $h_{t}^{c}$ is obtained by updating $u_{t}^{c}$.

**Fine-grained Graph Convolutional Network.** In Section 3.2, we present a batch-specific (task-specific) inter-variable dependency structure *A*. However, the fluctuation nature of the time-series data makes the inter-variable relationship change slightly at each timestamp, leading to diversity in temporal dynamics. In order to capture the slight changes, we calculate the cosine similarity between different variables at each timestamp. And the adjacency matrix is controlled within [0,1] through the ReLU activation function:(11)$$A_{ij}^{t}=1-ReLU\left(1-ReLU\left(A_{ij}+\cos\left({\tilde{z}}_{i}^{f,\psi },{\tilde{z}}_{j}^{f,\psi }\right)/\beta \right)\right),$$
where cos() is the cosine similarity function, and β is the perturbation parameter, which controls the perturbation strength. The slightly adjusted $A^{t}$ can be regarded as the fine-grained adjacency matrix at the *t*th timestamp, which is fed into the F-GCN to learn the fine-grained variable relationships at each timestamp and finally obtain the updated fine-grained feature vectors $h_{t}^{f}$:(12)$$h_{t}^{f}=\sigma \left({\tilde{D}}_{t}^{-\frac{1}{2}}{\tilde{A}}_{t}{\tilde{D}}_{t}^{-\frac{1}{2}}{\tilde{z}}_{t}^{f,\psi }W_{F}\right),$$
where $W_{F}$ is the trainable parameter, ${\tilde{A}}_{t}=A_{t}+I$ is the adjacency matrix of the self-recursive edges added at each timestamp, and ${\tilde{D}}_{t}$ is the degree matrix of ${\tilde{A}}_{t}$.

**Gated Aggregation Scheme.** Based on cross-view position-aware concatenation (i.e., the concatenation of the query set’s conditioning range data and task-specific contextual information extracted from the support set by the MV-PAN), the F-GCN and C-GRU capture temporal dynamics from two views, respectively. Then, we use a gated aggregation scheme to achieve appropriate information selection for generative predictions, thus generating the final prediction $h_{t}$:(13)$$\begin{array}{cc} \; & g_{f}=\sigma \left(W_{c}^{1}h_{t}^{c}+W_{f}^{1}h_{t}^{f}+b^{1}\right),\\ \; & g_{c}=\sigma \left(W_{c}^{2}h_{t}^{c}+W_{f}^{2}h_{t}^{f}+b^{2}\right),\\ \; & h_{t}=g_{f}*h_{t}^{f}+g_{c}*h_{t}^{c}, \end{array}$$
where $W_{c}^{1}$, $W_{2}^{c}$, $W_{f}^{1}$, $W_{2}^{f}$, $b^{1}$, and $b^{2}$ are trainable parameters, σ is the sigmoid activation function, and $g_{f}$ and $g_{c}$ are gating units. Finally, the vector $h_{t}\in R^{Fe}$ realizes the prediction of $\hat{y}$ through a trainable parameter $W_{y}\in R^{1\times Fe}$, i.e., ${\hat{y}}_{t}=W_{y}h_{t}$.

#### 3.3.4. End-to-End Stochastic Training

The training process learns shared parameters θ (i.e., the parameters of the MRI, MFE, and CV-CAG) across all tasks and parameters ϕ of the amortization networks (i.e., the parameters of the MV-PAN) through the loss optimization of the query set in different tasks. Given a new task, our model takes few-shot training datasets as inputs and then outputs a distribution over task-specific stochastic inputs ψ in a single forward pass, realizing fast and versatile learning and inference.

We layout the specific loss function for end-to-end stochastic training. First, ST-MeLaPI realizes learning via minimizing the expected value of KL divergence (between the approximate posterior predictive distribution and true distribution of the query set) averaged over the tasks: (14)$$\begin{array}{cc} \phi^{*} & =\underset{\phi }{argmin}E_{p(D)}[KL[p\left(\tilde{y}|D\right)\left|\right|q_{\phi }\left(\tilde{y}|D\right)]]\\ \; & =\underset{\phi }{argmax}E_{p(\tilde{y}|D)}\left[\log\;\int \int \;p(\tilde{y}|D,\psi^{s},\psi^{d})q_{\phi^{s}}\left(\psi^{s}|D^{c}\right)q_{\phi^{d}}\left(\psi^{d}|D^{f}\right)d\psi^{s}d\psi^{d}\right]. \end{array}$$

After introducing inputs $\tilde{Z}$ and shared parameters θ, we have
(15)$$\begin{array}{cc} L(\phi ) & =-E_{p(D,\tilde{Z},\tilde{y})}\left[\log q_{\phi }\left(\tilde{y}|\tilde{Z},\theta \right)\right]\\ \; & =-E_{p(D,\tilde{Z},\tilde{y})}\left[\log\;\int \int \;p(\tilde{y}|\tilde{Z},\psi^{s},\psi^{d},\theta )q_{\phi^{s}}\left(\psi^{s}|D^{c},\theta \right)q_{\phi^{d}}\left(\psi^{d}|D^{f},\theta \right)d\psi^{s}d\psi^{d}\right]. \end{array}$$

Finally, we leverage $L^{s/d}$ Monte Carlo samples to approximate the expectations of $\psi^{s}$ and $\psi^{d}$, formalizing the final end-to-end stochastic training loss function:(16)$$L\left(\theta ,\phi \right)=\frac{1}{M}\sum_{m=1}^{M}\log\frac{1}{L^{s}L^{d}}\sum_{l^{s}=1}^{L^{s}}\sum_{l^{d}=1}^{L^{d}}p({\tilde{y}}_{m}|{\tilde{Z}}_{m},\psi^{s},\psi^{d},\theta ).$$

## 4. Experimental Settings

### 4.1. Datasets

We conducted extensive experiments on three traffic benchmark datasets.

METR-LA: This is a traffic dataset collected by 207 loop detectors on highways in Los Angeles [46]. It contains data recorded every five minutes for 4 months between March 2012 and June 2012. The training/validation/testing data ratio was 7/1/2.

PEMS-BAY: This is a traffic dataset collected by the California Transportation Agency Performance Measurement System. In total, 325 sensors recorded data in aggregated 5 min intervals for 6 months. The training/validation/testing data ratio was 7/1/2.

EXPY-TKY: The EXPY-TKY dataset includes traffic speed information and corresponding accident reports for 1843 highways in Tokyo over three months (October to December 2021), with data collected every ten minutes. Consistent with the MegaCRN, we utilized the first two months (October and November 2021) as the training and validation datasets and the last month (December 2021) as the test dataset. Table 1 summarizes the specific spatio-temporal details of these datasets.

### 4.2. Experimental Details

#### 4.2.1. Baselines

Given the intrinsic dependence of traffic forecasting tasks on graph-structured relationships, our selection of baselines extends beyond the basic time series forecasting algorithm, Historical Average (HA), to include a focused comparison with recent graph-based time series forecasting methods. This includes representative spatio-temporal fusion algorithms such as the STGCN [38], DCRNN [39], and GW-Net [47]; Transformer-based forecasting algorithms like the STTN [48] and GMAN [49]; and models built upon the GCRN [50], including the CCRNN [51], GTS [9], and PM-MemNet [52]. Furthermore, we examined approaches based on meta-learning, specifically ST-GFSL [19] and the meta-graph-based MegaCRN [53].

#### 4.2.2. Hyperparameter Tuning

We employed the ADAM optimizer with a learning rate set to 0.0025 and a batch size of 16. We set the dimensionality of all feature vectors to 64. To ensure a certain degree of relevance within each batch, while simultaneously avoiding the premature exposure of test set labels through the training set in meta-learning, we randomized all samples in a specific pattern. For instance, when predicting future consecutive time points, samples were shuffled every 12 points, and after shuffling, every 16 samples were grouped into a batch, with the first four samples serving as the support set. On the METR-LA and PEMS-BAY traffic datasets, the model input length was fixed at 12. In the C-GRU, we set the coarse-grained receptive field to 10 and filled missing data with zeros. The fusion function, Ψ(), in this study was implemented using a concatenation operation. A generative forecasting approach was utilized; for example, in the METR-LA dataset, based on 12 consecutive time points, we predicted the next 6 consecutive time points by appending six zeros after the 12 time points. The positions of the appended zeros in the model’s final output represent our forecast results. We employed the MAE, RMSE, and MAPE as evaluation metrics. All models were trained/tested on an NVIDIA Tesla V100 32 G GPU.

#### 4.2.3. Evaluation Metrics

For the three traffic forecasting datasets, METR-LA, PEMS-BAY, and EXPY-TKY, we used three evaluation metrics, including the MAE, RMSE, and MAPE. The MAE, RMSE, and MAPE are calculated as follows:(17)$$\begin{array}{cc} \; & MAE=\frac{1}{T_{y}}\sum_{i=1}^{T_{y}}\left|\left(y_{i}-{\hat{y}}_{i}\right)\right|,\\ \; & RMSE=\sqrt{\frac{1}{T_{y}}\sum_{i=1}^{T_{y}}{\left(y_{i}-{\hat{y}}_{i}\right)}^{2}},\\ \; & MAPE=\frac{100\%}{T_{y}}\sum_{i=1}^{T_{y}}\left|\frac{y_{i}-{\hat{y}}_{i}}{y_{i}}\right|. \end{array}$$

### 4.3. Meta-Task Construction

**Sequential Construction.** An intuitive strategy is to sequentially split the data into different batches (tasks). In each task, the support set is formed by historical data, and the query set is formed by newly accumulated data, which is consistent with the training and inference requirement in time series prediction tasks. Note that there is no time-window overlap between the support set and query set in case of the information leakage problem. Specifically, based on the grid search method in hyperparameter learning, we set the batch size to 16, including 4 shots for the support set and 12 shots for query set. This was the default setting in our learning process.

**Generative Construction.** One drawback of the sequential construction strategy is the large time interval between the support set and the query set in the long-sequence prediction task. As a result, it easily generates inconsistent (even non-stationary) distributions, leading to a negative knowledge transfer issue. For example, in the task of predicting 12 future time points on the METR-LA and PEMS-BAY datasets, the interval between the support set and the query set will reach at least 60 min under our split method. In addition, the large time interval cannot be utilized as the conditioning window, thus failing to fully leverage all the historical data. To this end, we propose a generative strategy. An off-the-shelf time series prediction model can be pretrained to generate prediction window data to replace the original data. Hence, each conditioning window data can generate its own prediction window data without access to real data, thus bringing consistency between the support set and query set.

## 5. Results and Analysis

### 5.1. Traffic Flow Forecast

In the datasets for traffic flow prediction tasks, a large number of sensor nodes constitute a highly complex spatial network. This necessitates that ST-MeLaPI possesses strong capabilities for learning and capturing spatial features. Our model constructs its spatial structure from both macroscopic and microscopic perspectives to capture a richer set of structural features. As observed in Table 2, graph-structured algorithms significantly outperform traditional time series forecasting algorithms like the HA, validating the role of graph structures in multivariate time series prediction. Compared to the current advanced algorithm MegaCRN, on the METR-LA dataset, ST-MeLaPI achieved an average reduction of 16.3% in the MAE, 20.8% in the MAPE, and 32.5% in the RMSE. On the PEMS-BAY dataset, it showed an average reduction of 20.7% in the MAE, 27.4% in the MAPE, and 39.0% in the RMSE. On the EXPY-TKY dataset, there was an average decrease of 7.6% in the MAE, 5.9% in the MAPE, and 6.3% in the RMSE.

As a spatio-temporal meta-learning probabilistic inference method, ST-MeLaPI demonstrates rapid adaptability to complex and fluctuating multivariate time series forecasting scenarios. Compared with the latest meta-graph-based MegaCRN and the spatio-temporal meta-learning method ST-GFSL, ST-MeLaPI exhibits a superior performance. Utilizing the VERSA framework, ST-MeLaPI generates task-specific meta-parameters, circumventing the time-consuming second-order gradient computations found in methods like MAML, thereby enhancing the efficiency of meta-learning.

### 5.2. Ablation Studies

To further assess the effectiveness of each component within the model, we conducted ablation experiments on the PEMS-BAY dataset. The final results of these experiments are presented in Table 3.

Specifically, we first removed the MRI component from ST-MeLaPI (denoted as ST-MeLaPI_*MRI*−_) to validate whether the adjacency matrices captured via MRI effectively enhance the model performance in multivariate time series forecasting tasks. In ST-MeLaPI_*MRI*−_, a fully connected adjacency matrix was utilized as a substitute for the adjacency matrix generated via MRI. The results indicate that the dynamically generated adjacency matrices by MRI indeed improve prediction performance, especially evident within the traffic dataset (PEMS-BAY). Thus, the dynamic learning of inter-variable relationships is crucial in multivariate spatio-temporal dynamic analysis.

Subsequently, we removed structures related to the GCN (i.e., C-GCN and F-GCN) and GRU (i.e., C-GRU and F-GRU) from ST-MeLaPI to verify the necessity of multi-perspective learning. From a temporal perspective (ST-MeLaPI_*GCN*−_), the task-specific random input $\psi^{d}$ generated in the F-GRU from the support set was concatenated with the query set (control window data) and then inputted into the C-GRU for the final prediction. Similarly, from a spatial perspective (ST-MeLaPI_GRU−_), the task-specific random input $\psi^{s}$ generated by the C-GCN was added to the query set before being inputted into the F-GCN for the final prediction outcome. The significant performance decrease in single-view scenarios underscores the importance of considering both perspectives in the meta-learning framework for multivariate time series modeling.

Next, we removed MV-PAN from ST-MeLaPI (ST-MeLaPI_*MV*−*PAN*−_) to validate the effectiveness of the position-aware meta-learning strategy. As Table 3 shows, ST-MeLaPI_*MV*−*PAN*−_ performs even better than ST-MeLaPI in short-term forecasting (15 min). However, as the forecasting horizon extends (30 min, 60 min), the performance of ST-MeLaPI_*MV*−*PAN*−_ deteriorates, highlighting the advantages of ST-MeLaPI. We believe that longer sequence prediction tasks have more similar patterns that can be captured through the position-aware meta-learning strategy, demonstrating its necessity and effectiveness for long-sequence prediction tasks.

Finally, we removed the F-GCN and C-GRU from the CV-GAG in ST-MeLaPI to test the necessity of cross-view modeling. In the model withoutthe F-GCN (ST-MeLaPI_*F*−*GCN*−_), the final outcome was generated by the C-GRU. In the model without the C-GRU (ST-MeLaPI_*C*−*GRU*−_), the final outcome was generated by the F-GCN. As shown in Table 3, the cross-view information fusion strategy indeed significantly improves the performance, indicating that independent views lead to a decrease in information utilization.

Through comprehensive ablation experiments, we found that removing the GRU-related components used for temporal encoding significantly degraded the model performance. This result demonstrates the crucial role of capturing temporal features in traffic flow prediction tasks. Additionally, in a lateral comparison analyzing the impact of adjacency relationship modeling and graph convolution on the model, we observed that removing the MRI structure and using a fully connected matrix introduced more noise, significantly reducing model performance. This indicates that structured modeling is vital in reducing noise and enhancing model performance.

### 5.3. Few-Data Training

We conducted extensive experiments on the METR-LA dataset to validate our model’s learning capabilities in a low-resource learning environment. Specifically, we trained the model using only 30%, 20%, 10%, and 5% of the original training data. The final results are illustrated in Figure 5. For instance, ST-MeLaPI-10% indicates that only 10% of the original training data were utilized for multi-task meta-learning. We observed that ST-MeLaPI maintained a stable performance despite the reduction in the training data volume, and in many cases, models trained on a limited amount of data even outperformed those trained on the full dataset. Future research could focus on selecting the most informative time spans for training.

### 5.4. Dynamic Graph Generation

To explore the model’s capability in modeling multivariate relationships, this study employed a segment of multivariate time series data selected from the METR-LA dataset. Through the MRI module, a corresponding adjacency matrix was generated for in-depth visual analysis. Figure 6 displays the interconnectivity among 10 sensors captured during the learning process of the model. To further validate the practical utility of these adjacency relationships, we simulated a traffic accident scenario. Specifically, we mimicked the occurrence of a traffic accident by reducing the traffic flow at sensor 2 to observe its impact on the adjacency relationships. The analysis results clearly show that when traffic flow at an intersection decreases, the connection strength with other intersections weakens accordingly, thereby corroborating the effectiveness and accuracy of our model in understanding and depicting complex multivariate relationships.

### 5.5. Task Relevance Verification

The meta-learning paradigm assumes that tasks are both inherently related and stationary such that inductive transfer can improve sample efficiency and generalization. Otherwise, it easily leads to the negative knowledge transfer issue and catastrophic forgetting issue. Therefore, in this work, we split the sequential training data into different tasks, where each task spanned short time periods. As such, this generates a large number of meta-training tasks, bringing diversity as well as inherent relations. Since task-specific stochastic inputs (or parameters in a classification task) are generated based on support set for each task in ST-MeLaPI, we can compare the mean and variance between two tasks for a task relationship evaluation. In effect, we conducted the evaluation by analyzing the mean and variance of the spatial dependency-induced and temporal pattern-induced stochastic inputs for two consecutive learning tasks in the METR-LA dataset, respectively. As shown in Figure 7, the means and variances at each timestamp show strong similarities between the two consecutive batches and have some similarities. This provides insights for the effectiveness of the meta-learning framework.

## 6. Conclusions

In this study, we studied the multivariate time series prediction problem and proposed ST-MeLaPI to realize the efficient and versatile learning of complex temporal dynamics. Specifically, we designed a multiview position-aware spatial-temporal meta-learning probabilistic inference strategy, which essentially utilizes meta-learning to capture both shared and task-specific temporal dynamics. In particular, the multiview meta-learning of pairwise variable relationship-induced information and temporal pattern-induced view-specific information highlights the importance of sophisticated consideration for temporal dynamics-only and graph structure-only information. In addition, the proposed learning method realizes fast learning with forward passes trough inference networks without second-derivative computations during training, while harvesting diversity via stochastic neurons in each timestamp. The experiments on real-world data demonstrate the advantages of the model in multivariate time series prediction tasks.

## Figures and Tables

**Figure 1 sensors-24-04473-f001:**
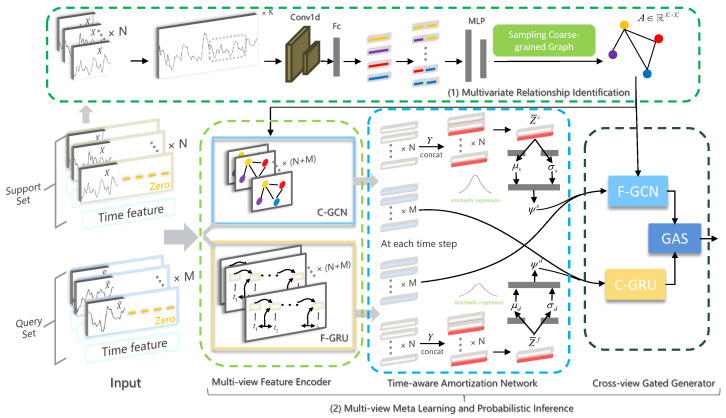
General overview of the ST-MeLaPI model framework. (1) Multivariate Relationship Identification (MRI) generates a probabilistic structure matrix through a convolution-based intra-variable temporal dynamic extractor and an inter-variable pairwise relationship predictor (MLP). The Gumbel-Softmax sampling technique is utilized to obtain an inter-variable pairwise relationship matrix *A*. (2) Multiview Meta-Learning and Probabilistic Inference (MV-MeLaPI). First, a multiview feature encoder (MFE) captures pairwise variable relationship-induced information and temporal pattern-induced view-specific information through a C-GCN and F-GRU, respectively. Note that only conditioning range data are fed into the encoder while prediction range data are replaced with zeros. Second, a multiview position-aware amortization network (MV-PAN) concatenates the view-specific output vectors with the corresponding prediction data of the support set, generating the task-specific stochastic input $\psi_{b}^{s}$ and $\psi_{b}^{d}$, at each timestamp. Third, a cross-view gated generator (CV-GAG) utilizes a cross-view mechanism that concatenates the view-specific feature vectors (generated through the F-GRU and C-GCN, respectively) of a query set with $\psi_{b}^{s}$ and $\psi_{b}^{d}$. Then, it further performs spatio-temporal representation learning using the F-GCN and C-GRU. Finally, a gated aggregation scheme (GAS) is used to achieve appropriate information selection for generative predictions.

**Figure 2 sensors-24-04473-f002:**
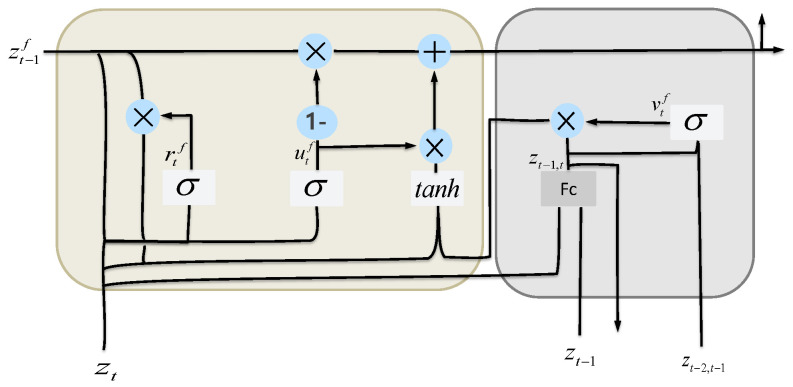
The structure of F−GRU.

**Figure 3 sensors-24-04473-f003:**
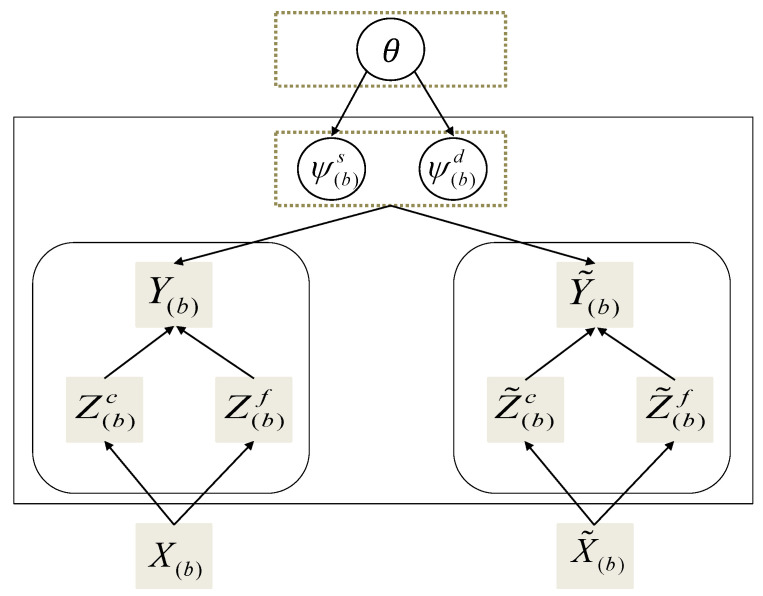
A directed graphical representation of the meta-learning and inference strategy.

**Figure 4 sensors-24-04473-f004:**
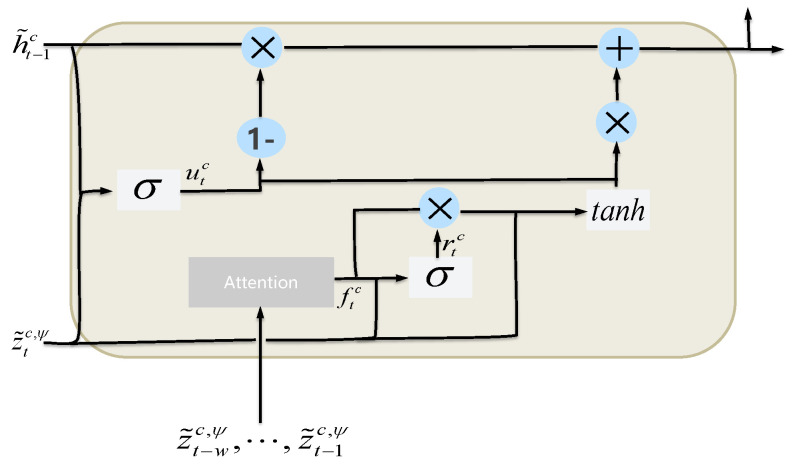
The structure of C-GRU.

**Figure 5 sensors-24-04473-f005:**
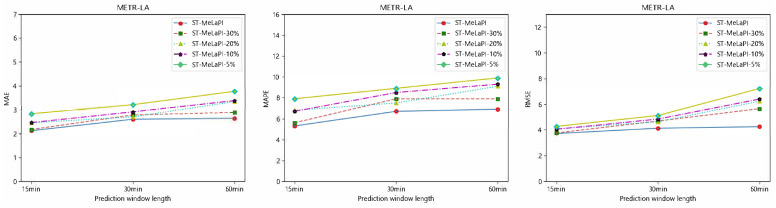
Model stability comparison using different amounts of training data on METR-LA dataset.

**Figure 6 sensors-24-04473-f006:**
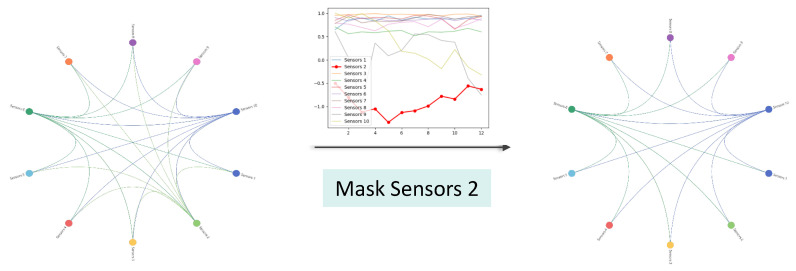
Connectivity pattern changes between sensors when sensor 2 experiences a sudden condition.

**Figure 7 sensors-24-04473-f007:**
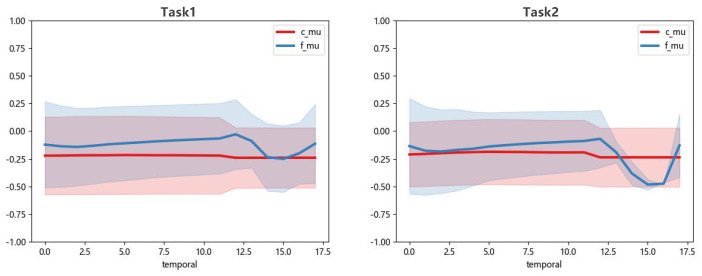
Distribution of the means and variances of the task-specific temporal pattern- and spatial dependency-induced stochastic inputs of the support set at each timestamp.

**Table 1 sensors-24-04473-t001:** Statistics of datasets.

Dataset	METR-LA	PEMS-BAY	EXPY-TKY
Start Time	1 March 2012	1 January 2017	1 October 2021
End Time	30 June 2012	31 May 2017	31 December 2021
Time Interval	5 min	5 min	10 min
Timesteps	34,272	52,116	13,248
Spatial Units	207 sensors	325 sensors	1843 road links

**Table 2 sensors-24-04473-t002:** Performance comparison on three traffic forecasting datasets.

Datasets	Models	15 min/Horizon 3			30 min/Horizon 6			60 min/Horizon 12		
		MAE	MAPE (%)	RMSE	MAE	MAPE (%)	RMSE	MAE	MAPE (%)	RMSE
METR-LA	HA [39]	4.16	13.0	7.80	4.16	13.0	7.80	4.16	13.0	7.80
	STGCN [38]	2.88	7.6	5.74	3.47	9.5	7.24	4.59	12.7	9.40
	DCRNN [39]	2.77	7.3	5.38	3.15	8.8	6.45	3.60	10.5	7.60
	GW-Net [47]	2.69	6.9	5.15	3.07	8.4	6.22	3.53	10.0	7.37
	STTN [48]	2.79	7.2	5.48	3.16	8.5	6.50	3.60	10.2	7.60
	GMAN [49]	2.80	7.4	5.55	3.12	8.7	6.49	3.44	10.1	7.35
	CCRNN [51]	2.85	7.5	5.54	3.24	8.9	6.54	3.73	10.6	7.65
	GTS [9]	2.65	6.8	5.20	3.05	8.3	6.22	3.47	9.8	7.29
	PM-MemNet [52]	2.65	7.0	5.29	3.03	8.4	6.29	3.46	10.0	7.29
	ST-GFSL [19]	2.90	N/A	5.59	3.56	N/A	6.96	N/A	N/A	N/A
	MegaCRN [53]	2.52	6.4	4.94	2.93	7.9	6.06	3.38	9.7	7.23
	ST-MeLaPI	2.12	5.3	3.72	2.60	6.7	4.13	2.64	6.9	4.25
	Generate-ST-MeLaPI	2.50	6.3	4.99	2.69	6.6	5.58	2.61	6.7	4.83
		15 min/horizon 3			30 min/horizon 6			60 min/horizon 12		
		MAE	MAPE (%)	RMSE	MAE	MAPE (%)	RMSE	MAE	MAPE (%)	RMSE
PEMS-BAY	HA [39]	2.88	6.8	5.59	2.88	6.8	5.59	2.88	6.8	5.59
	STGCN [38]	1.36	2.9	2.96	1.81	4.1	4.27	2.49	5.7	5.69
	DCRNN [39]	1.38	2.9	2.95	1.74	3.9	3.97	2.07	4.9	4.74
	GW-Net [47]	1.30	2.7	2.74	1.63	3.6	3.70	1.95	4.6	4.52
	STTN [48]	1.36	2.8	2.87	1.67	3.7	3.79	1.95	4.5	4.50
	GMAN [49]	1.35	2.8	2.90	1.65	3.7	3.82	1.92	4.5	4.49
	CCRNN [51]	1.38	2.9	2.90	1.74	3.9	3.87	2.07	4.8	4.65
	GTS [9]	1.34	2.8	2.84	1.67	3.7	3.83	1.98	4.5	4.56
	PM-MemNet [52]	1.34	2.8	2.82	1.65	3.7	3.76	1.95	4.5	4.49
	ST-GFSL [19]	1.56	N/A	3.18	2.07	N/A	4.58	N/A	N/A	N/A
	MegaCRN [53]	1.28	2.6	2.72	1.60	3.5	3.68	1.88	4.4	4.42
	ST-MeLaPI	1.08	2.2	1.81	1.21	2.6	2.24	1.46	3.0	2.44
	Generate-ST-MeLaPI	0.93	1.9	1.63	1.13	2.4	2.09	1.41	3.1	2.83
		10 min/horizon 1			30 min/horizon 3			60 min/horizon 6		
		MAE	MAPE (%)	RMSE	MAE	MAPE (%)	RMSE	MAE	MAPE (%)	RMSE
EXPY-TKY	HA [39]	7.63	31.2	11.96	7.63	31.2	11.96	7.63	31.2	11.96
	STGCN [38]	6.09	24.8	9.60	6.91	30.2	10.99	8.41	32.9	12.70
	DCRNN [39]	6.04	25.5	9.44	6.85	31.0	10.87	7.45	34.6	11.86
	GW-Net [47]	5.91	25.2	9.30	6.59	29.7	10.54	6.89	31.7	11.07
	STTN [48]	5.90	25.6	9.27	6.53	29.8	10.40	6.99	32.5	11.23
	GMAN [49]	6.09	26.5	9.49	6.64	30.1	10.55	7.05	32.9	11.28
	CCRNN [51]	5.90	24.5	9.29	6.68	29.9	10.77	7.11	32.5	11.56
	GTS [9]	N/A	N/A	N/A	N/A	N/A	N/A	N/A	N/A	N/A
	PM-MemNet [52]	5.94	25.1	9.25	6.52	28.9	10.42	6.87	31.2	11.14
	ST-GFSL [19]	N/A	N/A	N/A	N/A	N/A	N/A	N/A	N/A	N/A
	MegaCRN [53]	5.81	24.4	9.20	6.44	28.9	10.33	6.83	31.0	11.04
	ST-MeLaPI	5.02	22.3	8.41	5.89	26.1	9.45	6.77	31.3	10.83
	Generate-ST-MeLaPI	5.36	22.9	8.79	5.92	26.0	9.66	6.88	31.6	11.29

**Table 3 sensors-24-04473-t003:** Contribution of different components to ST-MeLaPI performance on PEMS-BAY dataset.

Datasets	Models	15 min			30 min			60 min		
		MAE	MAPE (%)	RMSE	MAE	MAPE (%)	RMSE	MAE	MAPE (%)	RMSE
PEMS-BAY	ST-MeLaPI	1.08	2.2	1.81	1.21	2.6	2.24	1.46	3.0	2.44
	ST-MeLaPI${\;}_{MRI-}$	1.78	3.7	2.87	2.15	4.9	3.55	3.18	8.6	5.41
	ST-MeLaPI${\;}_{GCN-}$	1.28	2.6	2.28	1.73	3.6	3.11	2.32	5.0	4.28
	ST-MeLaPI${\;}_{GRU-}$	3.08	7.3	4.86	3.14	8.1	5.19	4.70	13.4	7.26
	ST-MeLaPI${\;}_{MV-PAN-}$	0.88	1.7	1.73	1.23	2.6	2.51	1.80	4.1	3.52
	ST-MeLaPI${\;}_{C-GRU-}$	1.60	3.6	2.57	2.26	5.1	3.47	2.44	5.6	3.78
	ST-MeLaPI${\;}_{F-GCN-}$	1.42	2.9	2.45	1.52	3.1	2.53	1.61	3.3	2.60

## Data Availability

The data that support the findings of this study come from public resources.

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
