# Peer review of "Multiview Spatial-Temporal Meta-Learning for Multivariate Time Series Forecasting"

_sensors, 2024, doi:10.3390/s24144473_

Round 1

Reviewer 1 Report

Comments and Suggestions for Authors

Summary:

The paper proposes a spatial-temporal meta-learning probabilistic inference (ST-MeLaPI) framework for modeling the dynamics of multivariate time-series data. This framework assumes no prior knowledge about the dependencies of the time-series variables involved in the data and infers the relationships in a data-driven fashion. It then incorporates a multi-view strategy to capture and fuse the temporal and spatial information for better forecasting performance through task transfer. The paper benchmarks the proposed approach for the downstream task of traffic forecasting against representations of previous GNN-based, transformer-based, and meta-learning-based approaches to demonstrate its real-world utility.

Strengths:

  • The paper targets a real-world problem, where there exists no prior knowledge about the relationships between time-series variables and the data distributions evolve over time while we still want to capture and leverage the correlations between these variables for better forecasting.

  • The approach is benchmarked against a comprehensive set of classical, GNN-based, transformer-based, and meta-learning-based baselines and demonstrates superior performance in three real-world datasets. 

  • The ablation experiments justify the effectiveness of each proposed module and further experiments demonstrate how the learned patterns manifest in some real-world scenarios.

  Weaknesses:

  • The proposed MRI module to learn the inter-variable dependencies has multiple shortcomings, some of which have been already addressed in previous studies. In short, it learns an undirected, potentially unweighted graph structure without any assumption about prior relationships, which might not be desirable in many cases (see D1).

  • The writing can be confusing and hard to follow at times. Some sections lack a high-level description or intuition to help with understanding the explanation. (see D2)

  • The experiment setup is not very well specified and can also benefit from additional baseline comparisons. (See D3)

  • There are some formatting and citation issues in the paper. (See D4)

 Detailed Comments:

  • D1: There are multiple shortcomings with the proposed approach to learning the inter-variable structure:

    • The propose MRI module only focuses on the “structure” of the graph, e.g., a binary decision whether a relationship exists or not. However,  in many real-world applications, different time-series variables have different “degrees” of correlation rather than being either 100% correlated or not correlated at all, so a “soft” weight to characterize the relationships with a cut-off threshold to remove the weak relationship is more desirable and has been demonstrated in previous works to perform better. The authors are encouraged to check out these papers for reference:

      • Cao, Defu, et al. "Spectral temporal graph neural network for multivariate time-series forecasting." Advances in neural information processing systems 33 (2020): 17766-17778.

    • The work also assumes there exists zero prior knowledge about the relationships between the variables. While this is a very flexible assumption and enables the model to be applicable to a variety of domains, some level of “prior knowledge” often exists in real-world applications, and utilizing this information can benefit the effectiveness of the work to get rid of noisy relationships. This is demonstrated in this paper:

      • Hajisafi, Arash, et al. "Learning dynamic graphs from all contextual information for accurate point-of-interest visit forecasting." Proceedings of the 31st ACM International Conference on Advances in Geographic Information Systems. 2023.

    • The inferred graph structure is also “undirected”, while in the real-world downstream tasks, the correlations between different time-series variables are rather shown to be “directed”.

  • D2: The writing can become very hard to follow at times:

    • In the methodology section, the naming of the modules and reference to the appropriate modules can become very hard to follow (e.g., the authors introduce too many abbreviates such as "MV-PAN", "CV-GAG", "C-GCN", "F-GCN", etc.). Also, many terminologies are used that are likely invented by the authors (e.g., “coarse-grained” graph convolution, “fine-grained” GRU) which soon become confusing and make it hard to follow the bigger picture.

    • Some sections lack proper intuition or a higher-level explanation, e.g., why some ablation configurations in section 5.2 Intuitively degrade the performance of the model more than others? In section 5.3., it is mentioned the models trained on a limited amount of data even outperformed those trained on the full dataset. Why does this happen?

  • D3: The experiment setup and presentation can benefit from further clarification, e.g.,

    • How are the graphs for GNN baselines that are based on static graph structure constructed? Based on Euclidean distance or road-network distance or other metrics?

    • The hyper-parameter setup needs further information about the hyper-parameters and learning parameters to allow full reproducibility. E.g., the hidden dimensions of GRUs, graph convolution embedding dimension, etc.

    • The paper suggests a way to infer the graph structure, but to the best of the reviewer’s knowledge, none of the baselines used in comparison “infer” the graph structure. It is suggested to add at least one GNN-based work like the previously mentioned “StemGNN” that also infers the graph-structure for a more comprehensive comparison to similar studies.

    • Tables 3 and 4 can benefit from a row showing the relative “percentage improvement” or “performance drop” of the proposed model or ablated configuration for better readability.

  • D4: There exist some formatting issues:

    • DCRNN citations are repeated twice and cited separately throughout the text:

      • 38. Li, Y.; Yu, R.; Shahabi, C.; Liu, Y. Diffusion Convolutional Recurrent Neural Network: Data-Driven Traffic Forecasting. In Proceedings of the International Conference on Learning Representations (ICLR ’18), 2018.

      • 46. Li, Y.; Yu, R.; Shahabi, C.; Liu, Y. Diffusion convolutional recurrent neural network: Data-driven traffic forecasting. arXiv preprint arXiv:1707.01926 2017.

Please make sure to merge the duplicate citation entries.

  • The “conclusion” section is missing.

In summary, this paper presents a novel approach based on meta-learning to model the spatiotemporal dependencies in multivariate time-series data where data distribution evolves over time. To achieve this, the proposed spatial-temporal meta-learning probabilistic inference (ST-MeLaPI) framework first infers the relationships between time-series variables in a data-driven fashion. It then incorporates a multi-view strategy to capture and fuse the temporal and spatial information for better forecasting performance through task transfer. The paper benchmarks the proposed approach for the downstream task of traffic forecasting against baselines of previous GNN-based, transformer-based, and meta-learning-based approaches and demonstrates superior performance across three real-world traffic datasets. Overall, this is a very interesting work that can be valuable for researchers working on time-series modeling research. However, there are minor issues with the presentation and room for further improvements that the authors can incorporate.

Reviewer 2 Report

Comments and Suggestions for Authors

The paper introduces a novel ANN architecture for multivariate sensor data prediction (spatiotemporal urban traffic flows are used as an example). The essence of the proposed architecture is:

·         Multivariate Relationship Identification module – GTS, developed by Shang et. al.(2021)

·         MV-MeLaPI as a variant of VERSA (Gordon et. al., 2019)

·         Several minor improvements like fine-grained Gated Recurrent Unit Network module

The paper is well-written and describes all the necessary steps of ANN architecture development – a mathematical description of new modules, the proposed architecture, experimental comparison with state-of-the-art models, and ablation experiments. The only problem is a missing Conclusions section, which is necessary for highlighting the contribution.

In general, I don’t think that any new ANN architecture (from myriads of available combinations) is worth to be published in a scientific journal, but the empirical results for the traditional data sets (PEMS, METR, and EXPY) are really impressive.

Although source codes are not mandatory for Sensors publications, I highly recommend making them publicly available for better reproducibility (and as this is standard for contribution to machine learning).

Comments on the Quality of English Language

The paper is easy to read, so the language level is acceptable.
